# From a Lose–Lose to a Win–Win Situation: User-Friendly Biomass Models for *Acacia longifolia* to Aid Research, Management and Valorisation

**DOI:** 10.3390/plants11212865

**Published:** 2022-10-27

**Authors:** Florian Ulm, Mariana Estorninho, Joana Guedes de Jesus, Miguel Goden de Sousa Prado, Cristina Cruz, Cristina Máguas

**Affiliations:** 1cE3c–Center for Ecology, Evolution and Environmental Changes & CHANGE–Global Change and Sustainability Institute, Faculdade de Ciências da Universidade de Lisboa, Campo Grande, 1749-016 Lisbon, Portugal; 2Sousa Prado & Filhos, Agropecuária Lda, 7645-239 Vila Nova de Milfontes, Portugal

**Keywords:** *Acacia longifolia*, allometry, biomass models, invasive species, remote sensing, biomass valorisation

## Abstract

Woody invasive species pose a big threat to ecosystems worldwide. Among them, *Acacia longifolia* is especially aggressive, fundamentally changing ecosystem structure through massive biomass input. This biomass is rarely harvested for usage; thus, these plants constitute a nuisance for stakeholders who invest time and money for control without monetary return. Simultaneously, there is an increased effort to valorise its biomass, e.g., for compost, growth substrate or as biofuel. However, to incentivise *A. longifolia* harvest and usage, stakeholders need to be able to estimate what can be obtained from management actions. Thus, the total biomass and its quality (C/N ratio) need to be predicted to perform cost–benefit analyses for usage and determine the level of invasion that has already occurred. Here, we report allometric biomass models for major biomass pools, as well as give an overview of biomass quality. Subsequently, we derive a simplified volume-based model (BM ~ 6.297 + 0.982 × Vol; BM = total dry biomass and Vol = plant volume), which can be applied to remote sensing data or with in situ manual measurements. This toolkit will help local stakeholders, forest managers or municipalities to predict the impact and valorisation potential of this invasive species and could ultimately encourage its management.

## 1. Introduction

Almost all ecosystems worldwide are under intense pressure from invasive species, which constitute a fundamental driver of global change and biodiversity loss [1]. Among invasive plants, woody species, such as trees, are representing some of the most serious invaders worldwide [2] as they can act as autogenic ecosystem engineers, profoundly changing the invaded ecosystem through their own biomass [3]. Indeed, in general, invasive species biomass is a crucial determinant of the degree of invasion [4]. At the same time, aboveground biomass (AGB) plays a vital role in renewable energy transition [5] and carbon (C) reduction, either directly as renewable biofuel [6] or indirectly by binding C—for example, in compost [7]. Consequently, using invasive species AGB could create a win–win situation in which harvested biomass is valorised, while invasive species pressure on native vegetation and agricultural land is decreased.

Using invasive species as a biomass source has several advantages, as these plants need to be regularly cleared to manage their spread, they typically have high growth rates and are abundant [8] and have been shown to be useful as feedstock for biogas, biofuels, bioproducts and biorefining [9]. Important candidates for this use are *Acacia* spp. which are highly invasive in Mediterranean landscapes and there is little possibility for their total eradication, thus they will need to be managed and valorised in some way [10]. There are already several examples of their AGB use: exploitation of woody biomass for energy purposes [11] or as wood pellets for domestic boilers [12], as well as a vast array of biorefinery applications [13]. Another interesting usage of the whole plants, not just the woody parts, is the creation of compost from the chipped plant biomass [14]. This usage includes branches and foliage, which would be lost in most combustion applications. The first results are promising in that they show the potential use of *Acacia longifolia* biomass compost as organic amendments for agricultural purposes [15].

To make use of the biomass provided by invasive species, however, their location, abundance and total mass need to be modelled to provide stakeholders with estimates of the provided resources [8]. The most common methods to estimate biomass are combinations of remote sensing techniques and traditional allometric biomass models for AGB [16]. This combination is needed, as for the remote sensing techniques to yield data that are useful and of an acceptable certainty, local allometric models need to be developed [17]. Since their first use, allometric models have always been essential components for economic predictions, such as forest timber and biomass production; however, their significance is now further migrating into new areas of global importance, such as C accounting [18]. These models are based on the biological rule that all tree dimensions correlate with each other—for example, trunk diameter at breast height (DBH) and the whole AGB [16]. While allometric models can be very accurate for single tree species, they are also labour intensive as they always require plot- or plant-based destructive field sampling and weighing of the biomass [19], which is often only feasible for economically important species. Nevertheless, for invasive *Acacia* spp., several models were developed—for example, invasive *A. cyclops* and *A. saligna* in coastal areas of South Africa [20]—as well as for *A. dealbata* and *A. saligna* [21,22] in their respective native ranges in Australia. Conversely, for many very invasive species, no models are yet available, and a prime example is *Acacia longifolia*, even though it is a prolific invader in many ecosystems worldwide (see also Section 2.2).

However, it is not only the allometric model per se that is important if it is to be used in an applied setting, such as biomass recovery from management, but also the feasibility of measuring the input variables. For example, *A. longifolia* grows in a shrub-like fashion and constitutes a plant type exhibiting multiple stems, which make conventional models not practical for in-field application [23]. Other potential measures used in allometry are the total height of the plant (H) and its canopy size, which is approximated by an ellipse using the largest and the smallest canopy diameters (C_1_ and C_2_). These measures, joined with destructive harvesting of the whole individuum, are then used to create a volume-based biomass model for single plants, a family of models which have been successfully applied to various shrub species [24]. To upscale these models to the stand level, individual models are then coupled with density sampling to estimate stand AGB [23]. Another method is the direct measurements of stand variables, such as crown cover and total stand height [25], which can be combined to create a volume-based model [26]. The benefit of using volume-based models instead of diameter-based models is that they are easier to scale up using remote sensing methods, such as LIDAR or photogrammetry from unmanned aerial vehicles (UAVs) [27]. In particular, UAVs and photogrammetry have gained increasing traction in estimating AGB and plant volume in various settings due to its reduced costs, better scalability and higher data output [28].

Considerable effort has been invested into the development of new volume-based allometric models or merging existing allometric models with UAV data [29] but drawbacks remain, such as increased errors for smaller spatial scales [30] and for crown components (twigs, branches) compared to stems [31], which is problematic for carbon inventories and fuelwood estimations [32]. The work shown here aims to overcome these limitations by presenting a dual approach to biomass modelling, providing a variety of models for different settings, from single-tree manual measurements to stand-based measurements using UAVs and photogrammetry. On the basis of destructive harvesting for both cases, models were combined within the same framework and estimations can be given for each plant compartment, using either traditional variables, such as height and diameter, as well as UAV-derived variables for volume determination. Ultimately, the objective was to create modelling tools with different levels of time/resource investment to choose from, depending on the stakeholders’ context. As many biomass models already exist, potential research gaps on volumetric biomass models concerning *Acacia* spp. (and *A. longifolia* in particular) were elucidated through database mining. Individual and stand variables as well as biomass pools were described to provide context to the models presented here. In order to provide accurate sampling data on both total biomass and tree parts, exhaustive destructive biomass harvesting was combined with compartment separation (trunks and branches/foliage). Additionally, as an important objective of this study was the prediction of biomass quality, C, nitrogen (N) and phosphorous (P) of the compartments were determined in the laboratory to give estimates relevant for C sequestration potential and ratios of biomass. To create the final biomass models to be used, the most parsimonious and the most precise models were selected in both the manual single tree and the drone-based whole stand approach. Further simplification of the height measurement was provided to help stakeholders obtain a volume-based model without the need of a high resolution DTM. Lastly, models are discussed in various contexts to display the usefulness for both invasion management and biomass exploitation to turn double negative impacts for farmer and native vegetation into a potential win–win situation of biomass usage and invasive species management.

## 2. Results and Discussion

### 2.1. Database Search for Allometric Biomass Equations

To provide a basis for potential biomass equations to be employed on *Acacia longifolia*, the largest database on allometric biomass models available online, Globallometree [33], was searched automatically using R and regular expressions. In this search, 1596 entries for “Acacia” were found, of which 119 corresponded to the 12 *Acacia* species considered invasive in Portugal (Decreto-Lei no. 565/99, updated in Decreto-Lei nº 92/2019 to include all *Acacia* spp.). None of the reported equations was created in Europe and no equation was associated to *Acacia longifolia*. The equations used (88%) only one variable in most cases, which was stem diameter at breast height (DBH, 70.6%), at 10 cm (DB10, 14.3%), at 30 cm (DB30, 3.4%) or at ground height (DB0, 2.5%). Both height (H, 3.4%) and canopy diameter (CD, 2.5%) were very little represented. In 14 cases, two variables were employed, with 11 equations using DBH with H, 1 with CD with DBH, 1 with CD and D30 and 1 with DB0 with H. The final purpose of the allometric models presented here was to create volume-based biomass estimation; however, no equations using a combination of H and CD or any other type of plant volume-based equation were found. The most common (73.9%) equation structure was Y = a + b × ln(X), with Y being biomass and X being the explanatory variable, followed by power equations (24.4%), which are also synonymous with allometric equations if defined senso stricto [34].

### 2.2. Tree and Stand Measurements

For individual measurements, 37 trees were cut, exceeding the minimum number (n = 30) recommended for single, homogeneous stands [19]. For tree stands, 103 individuals were harvested, summing up to 119 stems in an area of ca. 275 m^2^ in total. Summary statistics for allometric and biomass related variables are shown in Table 1. Allometric measurements of stands and single trees were in similar ranges and thus comparable; however, some stands had larger volumes and stem diameters than single trees, which corresponded to higher overall biomass. Interestingly, while the largest stand had a trunk biomass only slightly larger than the largest singular tree (1.2 times), the same stand had more than twice (2.2 times) as much highest foliar and branch mass as the largest tree. This is linked to a different allocation pattern in singular tress opposed to stands or the dataset as a whole. Based on allometric biomass partitioning theory, which states that leaf mass should scale to the power of 0.75 to total plant volume (and mass) [35,36], lines were fitted to overall data, singular trees and stands (Figure 1a).

While the overall data fitted very well into the theoretical allometric relationship with a power of 0.73, singular trees exhibited rapidly a plateau in foliar and branch mass, deviating from this assumption. If expressed as a relation between the foliar and branch mass fraction (FBMF) relative to total biomass in singular trees, (Figure 1b), this plateau can be estimated using a segmented regression approach at 0.3, a value similar as reported earlier in a large-scale analysis for the leaf mass fraction in the same size class [37]. In stands, on the contrary, the FBMF is lower when we have small stands when compared to small singular trees, but then drops further linearly with increasing plant biomass, however, staying above the lowest levels found for single trees. A potential reason for this pattern could be the rapidly increasing allocation of plant resources to the trunk mass fraction (TMF, the reverse of the FBMF) as plants increase trunk growth to combat light competition with nearby individuals [38]. Furthermore, the rapid decline in the FBMF is linked to plant height (Pearson’s correlation: r^2^ = 0.46, *p* < 0.001), with individual trees being higher than trees in whole stands (Table 1 and Figure 1a,b) requiring them to proportionally investment more in stem growth vs. foliage for mechanical safety [37]. These observations fit well to established data that branch and foliar mass are more variable between species than stem or total aboveground biomass [39] and indeed, contrary to the foliar and branch biomass, a strong linear correlation between total aboveground biomass and trunk biomass was found (r^2^ > 0.98; *p* < 0.001). From a practical point of view, the data from destructive harvesting shown here concerning both the FBMF and the TMF are important as the FBMF is normally ignored in traditional biomass models that are concerned with timber production [18] and in the case presented here makes up roughly half of the biomass (FBFM_mean_ = 0.48; Figure 1b,c). While this biomass might not be merchantable in the traditional sense, it is an increasingly important logging residue [6] and crucial for fuelwood estimations and community forestry plantations [32] in developing countries.

While these observations are interesting from an allometric standpoint, they are also important from a practitioner’s point of view. The green waste from harvesting or eradication can be deviated either towards green waste compost or energy production, depending on its quality and proportion of woody and leafy material [7]. The FBMF is also the main determinant of the total C/N ratio of the biomass harvested (Figure 1c), highly correlated with height (Pearson’s correlation: r^2^ = 0.67, *p* < 0.001) and is the biomass with the lowest C/N ratio (Table 2). This ratio is especially important for the subsequent usage, because a C/N ratio between 20 and 30, as measured for the FBMF, is ideal for green waste composting [40]. On the contrary, high C/N ratios and low N content, as measured in the trunk (Table 2), are important if the biomass is used for other purposes, such as fuelwood, because N values should not exceed 0.6% to avoid NO_x_ emissions [41]. Therefore, using the data observed here, one could selectively harvest plants, using the FBMF for composting purposes and the trunk biomass for energy production. Whereas nutrient and C concentrations have a direct influence on the usage of the biomass, stable isotope ratios for C and N are important to understand plant and ecosystem functioning. In C_3_ plants, such as *A. longifolia*, δ^13^C can be used to gain insight into plant water use efficiency [42] and the plants observed here exhibit significantly depleted δ^13^C signatures in the foliar and branch fraction in stands, compared to single trees and trunk fractions. Combined with a negative correlation of these δ^13^C values with stand density (r^2^ = 0.37; *p* < 0.05), these observations suggest increased water stress in denser *A. longifolia* stands. This is in line with earlier works that describe this invasive species as a water spender [43] and might imply increased intraspecific competition as stands grow older. Contrary to δ^13^C, which is indirectly connected to water use efficiency, the N isotope ratio δ^15^N can be used to more directly to trace the source of N during acquisition. As a legume, *A. longifolia* can fix N directly from the atmosphere with the help of mutualistic bacteria, which translates into foliar δ^15^N signatures close to 0 (‰), similar to what is observed here and earlier works concerning *A. longifolia* [43,44]. Comparable to the δ^13^C signatures, the δ^15^N of foliar and branch fraction in stands was more depleted than in singular trees. The δ^15^N signature of branches and foliage was furthermore correlated with its N concentration (r^2^ = 0.22, *p* < 0.01), indicating that the signature, at least in part, traces N origin. While all values indicate atmospheric N fixation, the observed depletion could be related to an increased usage of N from other sources, e.g., soil N or symbiotic fungi [45].

### 2.3. Allometric Equations

Based on the database search (see Section 2.1), three equation types were selected for further analysis: linear models, linear models with ln-transformed independent variables and non-linear (power) models. For single trees, 27 separate equations and for stands 60 equations were fitted for the different plant biomass pools as well as the extrapolated total C, N and P (see Appendix A) using allometric variables shown in Table 1. Subsequently, for each dependent variable, equations with the lowest Bayesian Information Criterion (BIC) were selected (best model) and compared with the model that performed best with the smallest amount of variables used (most parsimonious possible) (Table 3 and Table 4). The BIC was chosen over other important criterions, such as the AIC (Akaike Information Criterion) to select for models with lower numbers of parameters [19]. For single trees, volume was the sole independent variable retained in all cases of the most parsimonious models, except for foliage, branches and both combined, being replaced by canopy area. This is in accordance with reports considering the significance of canopy area for allometric models in predicting tree crown mass that consists mainly of branches and foliage [46]. Trunk diameter, on the other hand, was a variable retained in the best models for total biomass, trunk biomass and branch biomass, stressing why diameter is of such importance in forestry allometry, which is mainly concerned with woody mass and, as this mass is the major C pool in trees, carbon markets [18].

In stands, the volume and the number of stems (density) were retained in the most parsimonious models for all variables (Table 4), highlighting the important direct or indirect effect of density through competition or tree social status on stand allometric models [47]. Apart from AGB pools, also models related to litter mass and SOM were elaborated. While these models might not have any commercial or production value, they can be important for predicting *A. longifolia* impact on native ecosystems. Both variables were highly linearity correlated (r^2^ = 0.72; *p* < 0.001), which again indicates litter accumulation as one of the drivers of SOM accumulation and changes underneath *A. longifolia* canopy [44]. Apart from these direct effects, modelling litter mass is also relevant for secondary ecosystem disservices from invasions, such as increased fuel loads that intensify wildfires [48].

### 2.4. Model Simplifications towards In Situ Application

While the allometric models shown in Section 2.3 are crucial to determine specific biomass pools, C, or nutrient masses, most of the variables used are difficult to obtain in the field without investing heavily in human resources. Plant volume, on the other hand, can be estimated over large scales using UAVs equipped with different types of sensors [28]. Thus, a simplified volumetric model would have the advantage of being easily applied over vast areas of stands if a DTM is available. Furthermore, in both single tree and stand cases, volume was also a variable always retained in the most parsimonious models for total biomass (Table 3 and Table 4), making this variable a good candidate for a combined model.

Combining volume data from stands and single trees, this model was fitted over the whole dataset (Figure 2a) and exhibits the biggest deviations for intermediate volumes (Figure 2b). The combined model had an RMSE of 13.47, which was 1.9 times higher than the RMSE of most parsimonious model of total biomass elaborated for single trees and 1.7 times higher for stands, respectively. Nevertheless, model performance was adequate for both small and large volumes. To validate this model in a realistic situation, a large section of the study area was chosen, the total plot volume of the cut *Acacia* stand (Figure 5 and Section 3.1) was determined in QGIS from the DTM (31,711 m^3^) and then used to predict total biomass to compare with actual biomass harvested. The total fresh biomass cut was 52,847 kg and dry weight was calculated at 31,869 kg using a water content of 60.31%, a value derived from the subsamples taken for total stands. The model predicted 31,141 kg with a 95% confidence interval between 28,079 and 34,203 kg, so model deviation from real data was only 728 kg, or 2.3%. This exercise contributes to evidence from a large scale, global database analysis of remote sensing data that proved height and canopy area to be good predictors of plant biomass, even without stem diameter [49]. Furthermore, the model developed here also fills a crucial gap in remote sensing of *A. longifolia* as an invasive species. Recently, progress has been made in *A. longifolia* detection by mapping this species with a UAV during the flowering stage [50] as well as by using neural networks for UAV-based automated detection [51]. The models presented here can build on these presence/absence models by estimating the impact in terms of biomass, a variable crucial for the determination of the invasion stage [4].

Despite the usefulness of the described model for in situ application by local stakeholders, it needs area and height as input to calculate the respective volume of the stand. Determination of area is straightforward in most cases, using publicly available satellite imagery. Height, on the other hand, must be determined either by hand, or from any available DTM. The model described here used a DTM of very high definition (2 cm/pixel), which might not always be available, thus, a minimum number of observations per area should be determined that still gives an accurate mean height measurement. This reasoning is based on an old rule of forestry, the Eichhorn’s rule, which states that a monospecific stands’ production is determined by its mean height [52]. To estimate this minimum number of height observations, an iterative procedure was used on the xyz point clouds from the DTM available. Different numbers of height (z) values were randomly selected from the point clouds, then the mean height calculated and correlated with total biomass (Figure 3). For each number of height measurements, this procedure was repeated 10,000 times and the mean r calculated to smooth prediction. Then, using segmented regression, a breakpoint was estimated at 18 observations per 100 m^2^, after which more height measurements do not significantly improve correlation with total biomass. Thus, for example, if one were to estimate mean height in a 100 m^2^ area, a pixel resolution of 20 m^2^ would be sufficiently accurate with 25 height measurements. Using this information, stakeholders can either make use of existing DTM data sources, or, in small-scale plots, measure random height points in an *A. longifolia* stand manually to obtain a decent biomass prediction value.

## 3. Materials and Methods

### 3.1. Field Measurements

All plants used for the work described here were sampled in an area adjacent to irrigated agricultural land classified as an arenosol (IUSS Working Group, WRB, 2006) in the perish of Longueira/Almograve, Odemira, Portugal (37°40′49.4′′ 8°45′49.9′′). The whole area had been clear cut 2008, followed by the rapid reestablishment of a monospecific *Acacia longifolia* stand. For the volumetric model of single plants, whole trees were collected at the 8, 9 and 12 February and from 19 to 21 June 2016. Before total destructive biomass harvest, diameters of trunks at soil height (D), the largest diameter (C_1_) and the perpendicular diameter (C_2_) of plant canopy, as well as total height (H) were measured for each plant (Figure 4). Canopy area (A) was then calculated as a circle from the mean of both canopy radii (C_1_ and C_2_) and volume calculated as a cone (1/3 × A × H). After cutting the trees at ground height, plant material was separated manually into foliage (phyllodes), branches, which were defined as woody material with diameter below 1 cm (“twigs”, following [53]) and trunks, defined as woody material with a diameter above 1 cm.

For the volumetric model based on photogrammetry, all trees in 11 square plots of ca. 25 m^2^ were collected on 13 and 14 October 2019. Before total destructive biomass harvest, all stems in the plots were counted (NrS) and their diameter at soil height measured to determine a mean diameter (D_mean_) per plot (Figure 4). Furthermore, one pooled soil sample (10 cm topsoil, with 3 subsamples) was taken per stand for soil organic matter (SOM) determination and a 1 m^2^ litter sample collected. Squares were delineated using a high-precision GPS (see Section 3.2). For stand biomass separation, biomass was harvested as described for individual trees; however, foliage and branches were sampled as a single pool (F + B). For both individual trees and stand biomass, each of the separated plant material was then weighed to determine wet weight with a spring scale (CR-300, Gram Precision, Barcelona, Spain) and subsequently subsamples were taken to determine dry weight. The final biomass harvest occurred between December 2019 and January 2020 in a total area of 8831 m^2^. In this case, biomass was harvested, chipped and then weighed together in a certified lorry weighing scale.

### 3.2. Drone Flight, GPS Points and Data Analysis

The drone flight was performed by the Terradrone company, using a SenseFly eBee (SenseFly, Cheseaux-sur-Lausanne, Switzerland) on 26 September 2019 on a clear-sky day. The flight altitude was 100 m, and the drone was equipped with two types of sensors: senseFly S.O.D.A. (SenseFly, Cheseaux-sur-Lausanne, Switzerland) and Sequoia Multispectral (Parrot, Paris, France). Using the sensor data, maps were created (Figure 5) with an image resolution of 2 cm per pixel and projected in the ETRS89 PT TM06 coordinate system with a total mapped area of 58,415 m^2^. A digital terrain model was created using structure from motion image processing techniques to create point clouds. Subsequently, GPS points were added to create 11 plots of ca. 5 × 5 m for later plant removal using an RTK: Leica GS10 high-precision receiver (Leica Geosystems AG, Hexagon, Stockholm, Sweden) and CHCNAV HCE320 data controller (Shanghai Huace Navigation Technology Ltd., Shanghai, China). GIS data analysis was performed using QGIS [54], area and volume was calculated per plot using the processing toolbox: Raster analysis > Raster surface volume. Volume was estimated from the whole digital terrain model; for area calculation, only pixels above 30 cm from the soil level were considered. As the whole study site was flat, the soil level was interpolated from nearby areas without vegetation (no further than 5 m from the harvested plots). The mean height per plot was calculated from z of the original point clouds (xyz file).

### 3.3. Sampling, Dry Weight Determination and Total P

Subsamples gathered in the field (see Section 3.1) for individual trees consisted of 50 phyllodes, 3 representative pieces of trunk material and 10 representative pieces of branches for each tree. In the case of stands, bags of ca. 20 litres of phyllode-branch material were harvested, as well as 5 representative trunk pieces. All subsamples were then dried at 70 °C until constant weight in a fan oven, following guidelines suggested by [53] to determine fresh weight/dry weight ratio. For SOM determination, soil samples were sieved with a 2 mm mesh sieve and then ignited in a muffle oven (600 °C, 6 h) to determine organic matter by loss on ignition. For total P of individual plant material, samples were ground to a fine powder in a ball mill (Retsch, Haan, Germany), ignited in a muffle oven (600 °C, 6 h) and subsequently acid-extracted (HCl, 1 M). Then, a malachite-green-based microscale method was employed as described by [55]. The colorimetric method was executed in 250 μL 96-well flat-bottom microtiter plates and analysed in a microplate reader (Rainbow, Tecan, Männedorf, Switzerland). For each single assay, a separate triplicate calibration curve was produced with KH_2_PO_4_ as a serial dilution in ultrapure water.

### 3.4. C/N Analysis

Dry organic matter (trunks, foliage and branches) was ground to a fine powder in a ball mill (Retsch, Haan, Germany). An amount of 5 ± 0.2 mg of the powder was packed into tin capsules. Stable isotope ratio analysis was performed at the Centro de Recursos em Isótopos Estáveis—Stable Isotopes and Instrumental Analysis Facility, at the Faculdade de Ciências, Universidade de Lisboa—Portugal. δ^13^C and δ^15^N in the samples were determined by continuous flow isotope mass spectrometry (CF-IRMS) [56], on a Sercon Hydra 20-22 (Sercon, UK) stable isotope ratio mass spectrometer, coupled to a EuroEA (EuroVector, Pavia, Italy) elemental analyser for online sample preparation by Dumas-combustion. Delta calculation was performed according to δ = [(R_sample_ − R_standard_)/R_standard_] × 1000, where R is the ratio between the heavier isotope and the lighter one. δ^15^N_Air_ values are referred to air and δ^13^C_VPDB_ values are referred to PDB (Pee Dee Belemnite). The reference materials used were USGS-25, USGS-35, BCR-657 and IAEA-CH7 [57]; the laboratory standard used was Wheat Flour Standard OAS/Isotope (Elemental Microanalysis, Okehampton, UK). Uncertainty of the isotope ratio analysis, calculated using values from 6 to 9 replicates of laboratory standard interspersed among samples in every batch of analysis, was ≤0.1‰. The major mass signals of N and C were used to calculate total N and C abundances, using Wheat Flour Standard OAS (Elemental Microanalysis, Okehampton, UK, with 1.47% N, 39.53% C) as elemental composition reference materials.

### 3.5. Statistical Analysis

Statistical analyses were performed with the package “stats” using version R 3.3.2 [58] and executed on Rstudio (IDE Version 1.3.959). For multiple comparisons with unequal sample size and non-normal data distribution (Table 2), a pairwise Wilcoxon rank sum test was performed. Figure 3 was created by using an algorithm that made use of the original point clouds (xyz file) from the digital terrain model derived by photogrammetry (see Section 3.2). The procedure randomly sampled heights (z) for each plot from n = 1 to 400 times per plot, calculated a mean of these heights and then created a linear model between mean height and harvested biomass to obtain Pearson’s r. To smooth out r, this was repeated 10,000 times and subsequently the mean r for each set of mean heights/biomass was plotted against the number of measurements. Lastly, a breakpoint regression was applied using the *segmented.lme()* function of the segmented package [59] to obtain the minimal amount of observations needed to get the highest resulting r. The breakpoint of the model (and models presented in Figure 1) was tested for significance using the Wald test with *anova.lme().*

The database search was performed on the Globallometree [40] global database using “Acacia” as the keyword for selecting the correspondent equations and entries. The dataset was then exported as a csv file and further analysed using R with regular expressions. Linear regressions were performed after verifying assumptions using the Breusch–Pagan Test for homoscedasticity and the Shapiro–Wilk Normality Test on the regression model residuals. Non-linear regressions (Figure 1 and allometric equations) were performed using the *nlsLM()* function from the minpack.lm package [60]. Allometric models were fitted and their BIC (Bayesian Information Criterion) was calculated. Then, for “best models”, the model with the lowest BIC was chosen while the model with the lowest BIC and the lowest number of variables was selected for “most parsimonious models”.

## 4. Conclusions

The objective of this work was to provide models for biomass and nutrient pools of the woody invasive *Acacia longifolia*. Ultimately, these models can help to map invasion impact, calculate biomass quantity and quality (C/N ratios, etc.) prior to harvest for a cost–benefit analysis or estimate fuel load and associated risks concerning wildfires. More complex models (Table 3 and Table 4) can be used by scientists and forest managers to ask questions requiring more specific answers, justifying the additional effort to measure all the necessary variables. However, the combined, volumetric model (Figure 2), on the other hand, may be considered a user-friendly equation that can be scaled up easily via remote sensing, thus providing a less accurate, but more convenient estimation. Lastly, making use of the high-precision DTM used in this work, an estimation of minimum height measurements per area was calculated (Figure 3), which could be crucial for local stakeholders who need to perform manual measurements or have data with low resolution available. In summary, this work provides a toolbox to model *A. longifolia* mass that can be adopted by a wide range of users and contributes to its control, either by providing necessary information for its valorisation or modelling its impact at a large scale.

## Figures and Tables

**Figure 1 plants-11-02865-f001:**
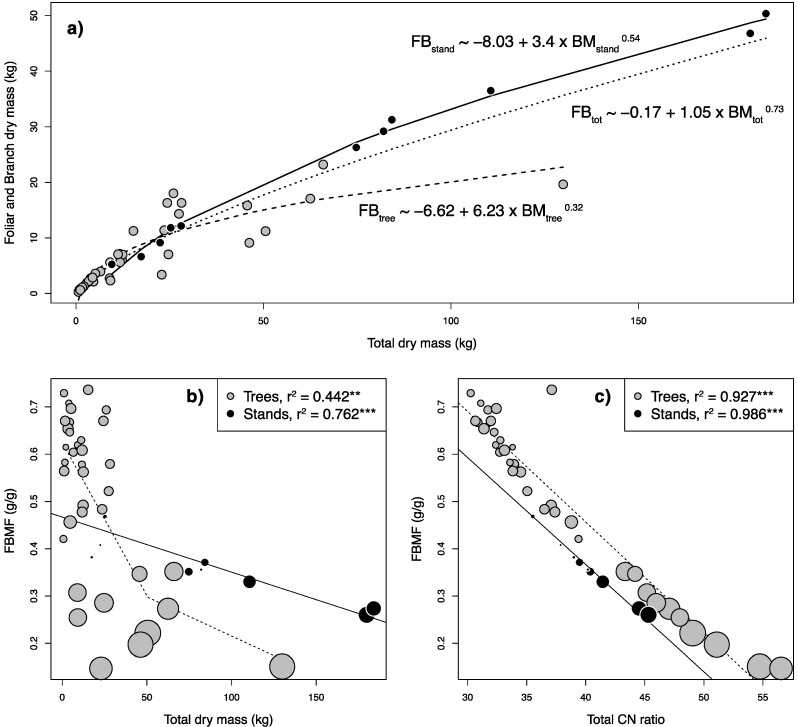
(**a**) Relationship between foliar and branch mass (FB) and total dry biomass (BM). Lines fitted follow the allometric relationship FB ~ a + b × BM^c^. (**b**) Regression between total biomass and the foliar and branch mass fraction of total biomass (FBMF). The correlation for stands is linear, while the correlation for trees is best described as a segmented regression after testing for a change in slope using Davies’ test (*p* < 0.05). The breakpoint was estimated at a FBMF of 0.3. Point size indicates plant height. (**c**) Regression between total C/N ratio and the foliar and branch mass fraction of total biomass (FBMF). Point size indicates plant height. ** = *p* < 0.01, *** = *p* < 0.001.

**Figure 2 plants-11-02865-f002:**
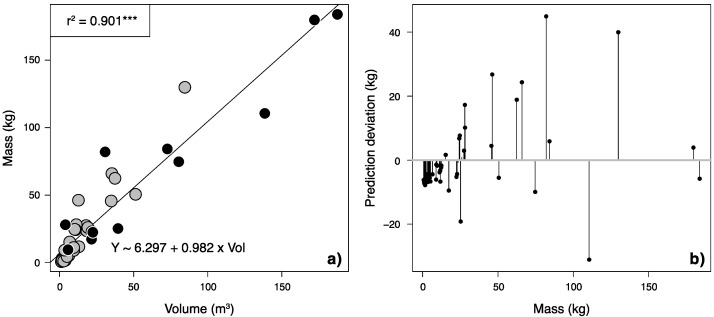
(**a**) The linear volumetric allometric model combining singular trees (grey dots) and stands (black dots). (**b**) Original masses plotted by the deviation of predicted values to original values. *** = *p* < 0.001.

**Figure 3 plants-11-02865-f003:**
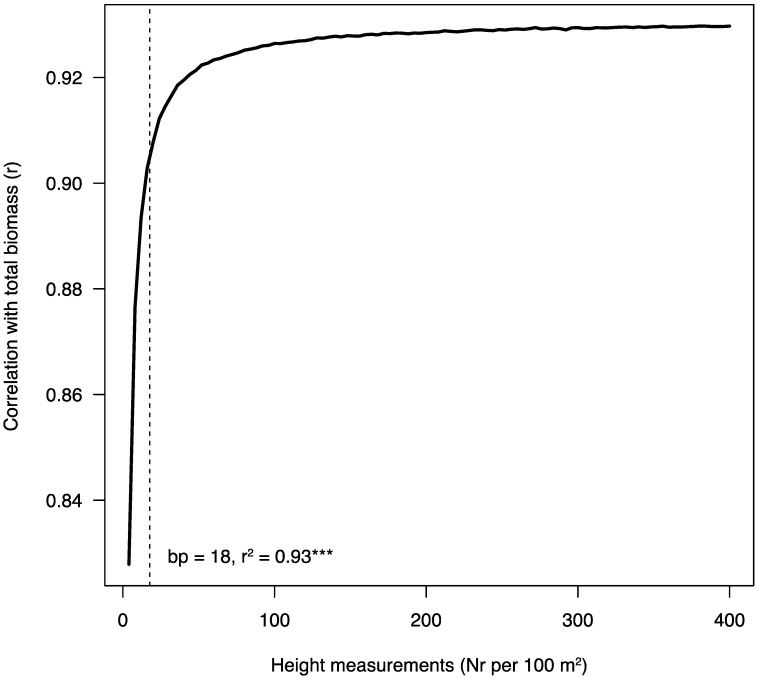
Correlation of the number of height measurements per 100 m^2^ with total biomass. Using segmented regression after testing for a change in slope with Davies’ test (*p* < 0.05), the point was estimated at 18. Nr = number; bp = breakpoint, *** = *p* < 0.001.

**Figure 4 plants-11-02865-f004:**
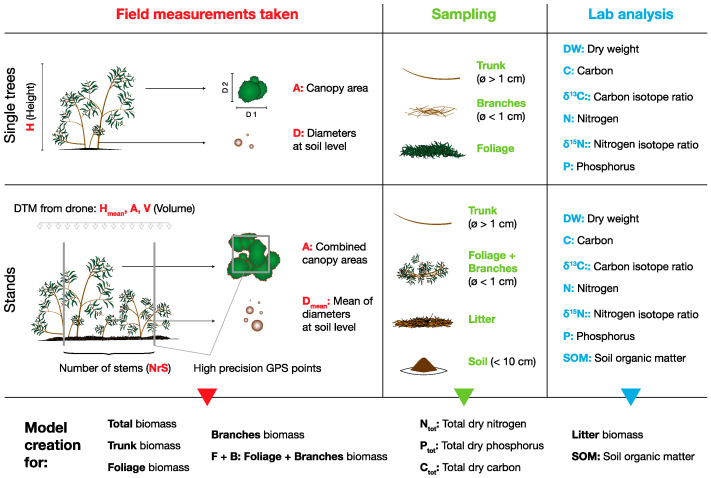
Overview of sampling scheme, summarising field measurements taken, sampled biomass pools and variables measured by lab analysis.

**Figure 5 plants-11-02865-f005:**
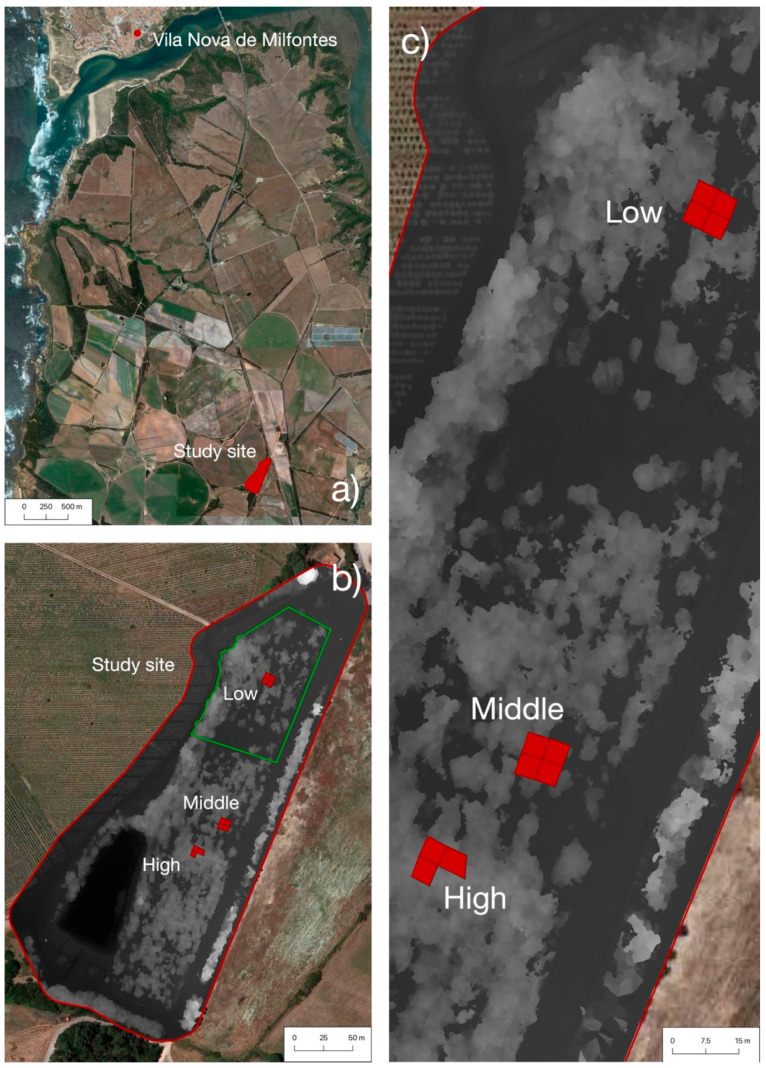
Zoom in on the study area. (**a**) Overview of the area surrounding the study site, which is situated in an agricultural area south of Vila Nova de Milfontes, Odemira, Portugal. (**b**) Close up of study site (marked in red), indicating the 3 different sampling sites (red boxes, density indicated in white letters) and the total biomass harvested (area outlined in green). (**c**) Zoom in on the sampling sites. Zoom (**b**,**c**) are with overlay of the digital terrain model (in tones of grey, darker grey indicates lower areas, and lighter grey higher areas).

**Table 1 plants-11-02865-t001:** Allometric and biomass related variables for single *Acacia longifolia* trees and whole stands (25 m^2^ clear cut quadrat). H = height; D = diameter at soil height; D_mean_ = calculated from all D per plot; H_mean_ = predicted from all pixels per plot (mean: 52,837 pixel per plot); SOM = soil organic matter; F + B = combined foliage and branches; n = 11 for stands, n = 37 for singular trees; ∆ = min-max difference (range).

	Variable	Min	Max	Mmean	∆
**Trees**	Volume (m^3^)	2.4	169.1	26.9	166.7
	Area (m^2^)	1.2	24.6	7.2	23.4
	H (m)	2.1	10.6	4.9	8.5
	D (cm)	1.9	25.5	9.8	23.6
	Trunks dry (kg)	0.3	110.3	12.5	110
	Branches dry (kg)	0.1	11.2	3.3	11.1
	Foliage dry (kg)	0.2	12	4.1	11.8
	F + B dry (kg)	0.3	23.2	7.3	22.9
	Total dry biomass (kg)	0.7	129.9	19.8	129.2
**Stands**	Volume (m^3^)	3.9	187.5	70.5	183.6
	Area (m^2^)	1.2	29.2	15	28
	Number of stems	3	45	10.8	42
	D_mean_ (cm)	7.2	34.1	19.9	26.9
	H_mean_ (m)	0.4	7	3	6.6
	Trunks dry (kg)	4.3	133.6	50.2	129.3
	F + B dry (kg)	5.2	50.4	24.1	45.2
	Total dry biomass (kg)	9.5	184	74.4	174.5
	SOM (%)	0.3	1.4	0.7	1.1
	Litter (kg/m^2^)	<0.01	1.9	0.7	1.9

**Table 2 plants-11-02865-t002:** Nutrient content, stoichiometric ratios and isotopic signatures of tissues collected for the biomass model. N = nitrogen, P = phosphorus, and C = carbon. The letters depict significant differences between the groups (pairwise Wilcoxon rank sum tests, *p* < 0.05); n = 11 for stands, n = 37 for singular trees. F + B = foliage and branches.

	Trees				Stand	
	Trunks	Branches	Foliage	F + B	Trunks	F + B
**C (%)**	43.2 (0.1) ^a^	45.1 (0.2) ^b^	48.1 (0.2) ^c^	47.7 (0.2) ^c^	44.1 (0.7) ^ab^	47.2 (0.5) ^c^
**N (%)**	0.6 (0.1) ^a^	1.1 (0.1) ^b^	2.3 (0.1) ^c^	2.2 (0.1) ^c^	0.6 (0) ^a^	2 (0.1) ^c^
**P (‰)**	0.8 (0.1) ^a^	-	2.4 (0.2) ^b^	-	-	-
**C/N ratio**	97.2 (10.5) ^ab^	42.3 (2.3) ^a^	21.7 (0.8) ^c^	24.2 (1.1) ^c^	72.2 (3.4) ^b^	23.5 (0.7) ^c^
**NP ratio**	11.3 (0.8) ^a^	-	11.4 (0.7) ^a^	-	-	-
**δ** ** ^15^ ** **N (‰)**	−1.2 (0.1) ^a^	−1.7 (0.1) ^ab^	−1.1 (0.1) ^a^	−1.2 (0.1) ^a^	−2.2 (0.1) ^c^	−2.1 (0.2) ^bc^
**δ** ** ^13^ ** **C (‰)**	−27.5 (0.2) ^a^	−25.9 (0.4) ^b^	−28.4 (0.2) ^ac^	−28.1 (0.3) ^ac^	−27.5 (0.2) ^ab^	−29 (0.4) ^c^

**Table 3 plants-11-02865-t003:** Most parsimonious and best fit models for various dependent variables of single trees. H = height (m), A = area (m^2^), V = volume (m^3^), and D = diameter at soil height (cm). RMSE = root mean square error, C_tot_ = total carbon, N_tot_ = total nitrogen, P_tot_ = total phosphorus, and F + B = foliage and branches. Dry mass is in kg.

	Most Parsimonious Model		Best Model		
Y (Dry Mass)	Equation	RMSE	Equation	RMSE	ΔΡΜΣΕ (%)
**Total**	Y ~ 0.25 + 1.46 × V	7.93	Y ~ −7.56 + 1.3 × D + 1.12 × V	7.21	9.9
**Trunk**	Y ~ 0.73 + 0.34 × V^1.29^	6.28	Y ~ −4.5 + 1.4 × 10^−05^ × D^4.78^ + 0.06 × H^2.61^ + 1.92 × A^0.64^	3.5	79.6
**Foliage**	Y ~ −2.48 + 3.83 × ln(A)	1.9	Y ~ −2.48 +	1.9	0
			3.83 × ln(A)		
**Branches**	Y ~ −1.03 + 0.23 × D + 0.29 × A	1.55	Y ~ −1.03 + 0.23 × D + 0.29 × A	1.55	0
**F + B**	Y ~ 0.2 + 0.99 × A	3.28	Y ~ 0.2 + 0.99 × A	3.28	0
**C_tot_**	Y ~ 0.25 + 0.64 × V	3.51	Y ~ −3.27 + 0.57 × D + 0.49 × V	3.18	10.3
**N_tot_**	Y ~ −0.07 + 0.06 × V^0.64^	0.08	Y ~ −0.05 + 0.02 × D + 0.01 × V	0.08	2.5
**P_tot_**	Y ~ 0.003 + 0.001 × V	0.01	Y ~ −0.01 + 0.0013 × D + 0.001 × V	0.007	14.3

**Table 4 plants-11-02865-t004:** Most parsimonious and best fit models for various dependent variables of stands. H_mean_ = mean height (m), A = area (m^2^), V = volume (m^3^), and D_mean_ = mean diameter at soil height (cm). RMSE = root mean square error, C_tot_ = total carbon, N_tot_ = total nitrogen, P_tot_ = total phosphorus, NrS = number of stems, SOM = soil organic matter, and F + B = foliage and branches. Dry mass is in kg.

	Most Parsimonious Model		Best Model		
Y (Dry Mass)	Equation	RMSE	Equation	RMSE	ΔΡΜΣΕ (%)
**Total**	Y ~ −31.01 + 0.22 × V^1.28^ + 19.96 × NrS^0.41^	7.23	Y ~ −190.97 + 29.28 × NrS^0.41^ + 0.0008 × A^3.38^ + 0.013 × H_mean_^4.51^ + 78.16 × D_mean_^0.24^	2.13	239.5
**Trunk**	Y ~ −28.52 + 0.07 × V^1.43^ + 18.15 × NrS^0.36^	5.78	Y ~ −109.35 + 21.64 × NrS^0.4^ + 7.7 × 10−^05^ × A^4^ + 0.001 × H_mean_^5.52^ + 30.95 × D_mean_^0.34^	0.9	539.9
**F + B**	Y ~ 2.13 + 0.24 × V + 0.46 × NrS	2.67	Y ~ 2.13 + 0.24 × V + 0.46 × NrS	2.67	0
**Litter**	Y ~ −0.28 + 0.02 × V + 0.06 × NrS	0.63	Y ~ −0.81 + 0.001 × H_mean_^4.13^ + 0.58 × NrS^0.48^	0.32	97.5
**C** ** _tot_ **	Y ~ −13.79 + 0.1 × V^1.27^ + 8.89 × NrS^0.41^	3.24	Y ~ −107.24 + 12.9 × NrS^0.42^ + 0.0004 × A^3.36^ + 0.007 × H_mean_^4.38^ + 54.39 × D_mean_^0.18^	0.94	244
**N** ** _tot_ **	Y ~ −0.02 + 0.01 × V+ 0.02 × NrS	0.09	Y ~ −2.94 + 0.0002 × V^1.67^ + 1.29 × D_mean_^0.2^ + 0.51 × NrS^0.34^	0.04	154.3
**SOM**	Y ~ 0.42 + 0.003 × V	0.16	Y ~ 0.13 + 0.0144 × NrS − 0.038 × A + 0.18 × H_mean_ + 0.019 × D_mean_	0.09	67.7

## Data Availability

For review of published biomass models: http://globallometree.org/ (accessed on 20 October 2022).

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
