# Peer review of "From a Lose–Lose to a Win–Win Situation: User-Friendly Biomass Models for Acacia longifolia to Aid Research, Management and Valorisation"

_plants, 2022, doi:10.3390/plants11212865_

Round 1

Reviewer 1 Report

From lose-lose to win-win: Use-friendly biomass models for Acacia longifolia to aid research, management and valorisation.

This is an interesting and valuable article, in which the authors develop a set of biomass models for Acacia longifolia an aggressive invasive species which strongly affect community structure, species composition and nutrient cycling in invaded ecosystems.

With these models, the authors try to help stakeholders in order they can easily manipulate plant biomass for different purposes, as compost, substrate, or biofuel.

The abstract and keywords are well written, and the authors clearly explain which are the main topics, results, and conclusions of the study.

Introduction

The introduction is well written, the authors clearly explain some of the problems of invasive species and the need to obtain models to estimate plant biomass.

Lines 93-108. In the last paragraph of introduction, the authors explain the sequence of their work, in order to obtain the different models. I think, it should be clearer if they summarise this sequence of the work in the different objectives of the study.

Material and Methods

Figure 1. I would appreciate to improve the quality of figure 1. Although it is a sampling schematic plot, some of the information is difficult to understand.

In the same way, the quality of Figure 2 is very bad. Plot (a) which shows an aerial view of the study sites is clear, but, and I can hardly distinguish the study sites marked in red in plot (b) and in plot (c) I can’t understand almost anything.

The rest of material and methods and statistical analysis are well explained, and I have no questions on them.

Results and Discussion

I consider that the section of results and discussion is appropriate, with the presentation of all the morphological data and the production and discussion of all the models and their application.

I only have a couple of questions. In Table 2, the authors exhibit a series of data of nutrient content, stoichiometric ratios and isotopic signatures of plants collected for the biomass model, but the isotopic data are not analysed in the text. If these data are shown in the table, I would appreciate that they will be mentioned and analysed in the text.

On the other hand, the quality of figure 4 is not very good, and in figure 4a), it is very difficult to distinguish between individual trees and stands.

Finally, the conclusions are clear and are well presented.

In my opinion, this manuscript deserves to be publish in the journal after minor revision.

Reviewer 2 Report

The paper needs some minor corrections. The model and the approach.,e.g., using a drone has been described and is relatively clear however the authors need to expand on the applications of the model that they have described, and establish connections to the economic and sustainability impacts for different regions. I believe the So What aspect of the paper needs to be better reflected.
